# Implicit Differentiable Outlier Detection Enables Robust Deep Multimodal Analysis

**Zhu Wang**     **Sourav Medya**     **Sathya N. Ravi**
Department of Computer Science, University of Illinois at Chicago
{zwang260,medya,sathya}@uic.edu

## Abstract

Deep network models are often purely inductive during both training and inference on unseen data. When these models are used for prediction, but they may fail to capture important semantic information and implicit dependencies within datasets. Recent advancements have shown that combining multiple modalities in large-scale vision and language settings can improve understanding and generalization performance. However, as the model size increases, fine-tuning and deployment become computationally expensive, even for a small number of downstream tasks. Moreover, it is still unclear how domain or prior modal knowledge can be specified in a backpropagation friendly manner, especially in large-scale and noisy settings. To address these challenges, we propose a simplified alternative of combining features from pretrained deep networks and freely available semantic explicit knowledge. In order to remove irrelevant explicit knowledge that does not correspond well to the images, we introduce an *implicit Differentiable* Out-of-Distribution (OOD) detection layer. This layer addresses outlier detection by solving for fixed points of a differentiable function and using the last iterate of fixed point solver to backpropagate. In practice, we apply our model on several vision and language downstream tasks including visual question answering, visual reasoning, and image-text retrieval on different datasets. Our experiments show that it is possible to design models that perform similarly to state-of-the-art results but with significantly fewer samples and less training time. Our models and code are available here: `https://github.com/ellenzhuwang/implicit_vkood`

## 1 Introduction

Numerous neural network models are constructed via the stacking of explicit layers that transform inputs into outputs through a sequence of operations associated with the designated layers. Although these explicit layers are expressive, they may be unnecessary in many large-scale applications where only the function value and its gradients are required. For instance, it is possible to employ implicit layers for diverse tasks, including hyperparameter optimization [7], meta learning [50], and solving inverse problems in image processing [21]. Indeed, if a desired input-output correspondence *within* a network can be expressed as an optimization problem, gradients can be computed efficiently [30].

Recently, JFB [17] proposed an efficient method to backpropagate through implicit layers by implementing them as a Network Operator $\mathcal{N}(\cdot)$. Here, $\mathcal{N}$ is defined as a sequential application of a map $F(\cdot)$, which is guaranteed to converge in $T < \infty$ iterations. Formally, $\mathcal{N}(x)$ can be represented as,

$$\mathcal{N}(x; \Theta) = \underbrace{F \circ F \circ \cdots \circ F}_{T \text{ times}}(x; \Theta),\tag{1}$$

where $\Theta$ corresponds to parameters of a potential multimodal model. The convenience of $\mathcal{N}$ in Equation (1) lies in its ability to backpropagate through $\mathcal{N}$ without requiring its full sequence (or

37th Conference on Neural Information Processing Systems (NeurIPS 2023).

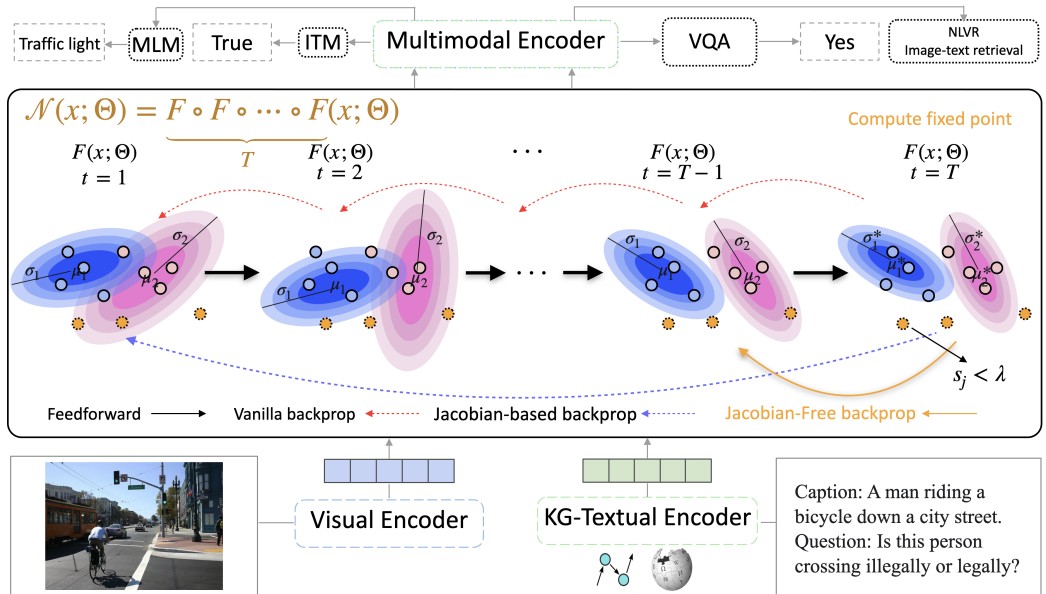

Figure 1: Our Proposed Architecture. The inputs are images and captions/questions. Image patches are processing by a visual encoder, while the caption, integrated with a knowledge graph, is passed to a language encoder. The extracted features are then fed into the OOD detection layer. This implicit layer identifies noise concepts before forwarding the features to the multimodal encoder. The ITM and MLM are training objectives, and VQA, NLVR, and image-text retrieval are downstream tasks.

trajectory) or solving an inverse problem with its Jacobian matrix for gradients. In Vision tasks, implicit maps like clustering with fixed parameters – for instance, $\mathcal{N}$ could be $k-$means with fixed means – have been shown to enhance performance [57]. To enable small-scale custom applications or to fine-tune large-scale models, fixed parameters are insufficient, so we consider the problem of evaluating gradients with respect to the means efficiently.

**Introducing "Benign" noise in Multimodal Pipelines.** Recent advancements [65, 27] have demonstrated that the combinations of multiple modalities improve understanding and boost generalization performance. However, these models, trained by a simple relation (*matched or unmatched*) between the image and text pair, often falter in capturing important semantic information and implicit dependencies within datasets. For example, as depicted in Figure 1, these models may recognize the green light but fail to infer the legality of a person crossing at the green light. Thus, the incorporation of "noisy" external knowledge presents a potential solution. In this work, we explore the design of an implicit Network Operator $\mathcal{N}$ to manage feature-level noise during training. Specifically, $\mathcal{N}$ is employed to integrate external knowledge in multimodal learning, which is a deep network with parameters $\Theta$ developed to align multiple modalities [49, 46].

**Using External Knowledge for Reasoning Purposes.** A knowledge graph collects information, organizes it in a specific structure, i.e., ontology, and is utilized to deduce novel insights and knowledge from the integrated information [16]. Often, external knowledge may be beneficial for reasoning tasks [22]. Most recent knowledge-based multimodal models [22, 36, 53] focus on entity retrieval methods from external knowledge resources such as structural knowledge graphs and large language models. Unfortunately, these methods become impractical when combined with features derived from external knowledge resources that are *prone* to incompleteness and noise. Furthermore, vision-language models like those in [18, 66] often struggle to filter noise pairs during training, resulting in slow convergence. For instance, in Figure 1, given an object "street" and relation "locatedAt", noisy objects like "fire hydrant" are returned with high confidence by ConceptNet [55]. However, the "fire hydrant" is absent in the input image, which is undesirable for training objectives, such as image-text matching .

We propose an *implicit out-of-distribution (OOD) detection layer* to handle outliers from external knowledge as discussed above. We demonstrate how such a layer can be expressed as a network operator $\mathcal{N}$ in Equation (1) for memory saving and efficient training. Specifically, in the ongoing example in Figure 1, our proposed approach approximates the density of in-distribution features,

which are extracted from the triplets in the caption (question) corpus, by a fixed point iterative layer. Thus, $F$ is trained with implicit differentiation at the fixed point according to OOD scores [44] for the retrieved knowledge triplets. Overall, we provide an end-to-end training framework with knowledge graphs and multimodal models in an implicitly differentiable manner, as illustrated in Figure 1 showcase the effectiveness of the proposed implicit layer based on $\mathcal{N}$ in large-scale multimodal pipelines. We conducted several experiments on multiple vision and language tasks, such as visual question answering and image text retrieval. We have included extensive evaluations and ablation studies to demonstrate the improvements in memory usage and training time compared to vanilla backpropagation.

**We summarize our three primary contributions as follows:** **(1)** Our proposed implicit layer can perform outlier detection tasks efficiently with less memory costs during backpropagation in practical settings. This layer can scale to different datasets and multimodal backbones. **(2)** We demonstrated that integrating explicit knowledge enhances multimodal fusion models. Our model learned from visual and textual modalities while using external knowledge seamlessly that may available in certain situations. **(3)** Resulting model after training and/or fine-tuning, outperforms baseline models on various downstream tasks. Moreover, we provided various ablation studies and an interactive simulation based user study, which shows that users can sustain their desired in-distribution inputs or features while being able to update model parameters simultaneously.

## 2   Related Work

**Deep implicit layers.** Implicit differentiation has been an emerging and competitive alternative to explicit unrolling for backpropagation. From the practical point of view, it is often memory-efficient and numerically stable compared to vanilla unrolling [7, 25]. Recent works apply implicit functions and layers in various problems with deep neural network, such as optimization layers [2, 6], convolutional sparse modeling [33] and object representations [9]. Moreover, implicit layers can be used to produce representations that are robust to perturbations, as well as better empirical performance in downstream predictions. For example, [3, 5] used path independence as a technique to improve the OOD generalization. The main argument is that path independence can be guaranteed while training using implicit layers instead of unrolling which is crucial for our applications.

**Out-of-distribution (OOD) detection.** Out-of-distribution detection is usually studied from the statistical point of view, mainly under Robust Statistics [26]. We will summarize recent developments focusing on our use cases. For OOD detection, Maximum Mahalanobis Distance [29] and energy score [37] are statistical methods for detecting features that may not be obtained using training data. Recently, GEM [44] has introduced a provable and competitive OOD detection method to estimate outliers for deep networks. GEM score can be used to recognize irrelevant distributions of concepts during training. The primary focus is on the detection performance with simulated outliers, whereas our focus in this paper is on utilizing them in real-world noisy data for *training* purposes efficiently.

**Vision-and-language transformers and knowledge-based VQA.** Most recent vision-and-language models have shown improved performance by optimizing self-supervision based losses. For example, [10, 27, 32, 31, 54, 1] introduce multimodal architectures wherein cross-modal features are used to learn combined representations of visual and textual contents that can be used to improve predictions. In Vision and Language Processing, knowledge representations in the form of extracted consensus or semantic concepts are aligned with visual concepts to construct concepts vocabulary [24, 19, 60]. In knowledge-based VQA, different explicit knowledge bases can be integrated for answering purposes [40, 63, 23, 22, 53, 36, 11]. In this work, we explore the effectiveness of utilizing external knowledge in deep multimodal models during training.

## 3   Implict OOD Detection Layer for Multimodal Analysis

**Basic Notations.** We denote input features as $x_i \in \mathbb{R}^d, i = 1, \dots, N$ including features of extracted knowledge triplets $l_j \in \mathbb{R}^d, j = 1, \dots, M$. The goal of OOD layer is to compute a score $s(l_j) \in \mathbb{R}_+$ for $l_j$ to estimate density of in-distribution (ID) data. In this section, we explain how to estimate the distribution of the ID features as a (normalized) linear combination of simpler distributions using an implicit layer to detect outliers and backpropagate for gradients, efficiently.

## 3.1 Finding Fixed Points for Forward Pass

Finite mixtures with $k$ components are conceptually simple, and computationally attractive. Mathematically, they can produce accurate approximations to most density functions [43]. So, we approximated the density of ID features using a Gaussian Mixture Model (GMM) and used $\mu_k^* \in \mathbb{R}^d, \sigma_k^* \in \mathbb{R}^{d \times d}, k = 1, \ldots, K$ to denote the optimal means and covariance matrices of $K$ components.

To obtain an optimal GMM, we solved $\mu^*$ and $\sigma^*$ by using Expectation Maximization (EM) algorithm [45]. Our main observation is that the update rule used in an EM-based algorithm of $\mu_k$ on the current iterate $t \in T$ can be written as a fixed point iteration as follows:

$$\mu_k^{t+1} \leftarrow \frac{\sum_{i=1}^{N} \exp(-w(\mu_k^t)) x_i}{\sum_{i=1}^{N} \exp(-w(\mu_k^t))} \tag{2}$$

where the weight of current iterate $\mu_k^t$ on $x_i$ denoted by $w(\mu_k^t) := \sum_{k=1}^{K} \|\sigma_k^{-0.5}(x_i - \mu_k^t)\|_2^2$. We updated $\sigma_k$ similarly using $(x_i - \mu_k^t)(x_i - \mu_k^t)^T$ in the numerator of Equation (2). Therefore, we obtained an approximation of in-distribution density that can be used for further outlier detection.

## 3.2 GEM score for Outlier Detection

After computing the optimal means $\mu^*$ and covariances $\sigma^*$ using the above described fixed point function, the final ingredient we need is a samplewise and memory efficient forward pass OOD detection method. After obtaining the ID parameters, we can compute a score for each $l_j$, the features derived from external knowledge triplet. In deep network context, the recently introduced GEM score [44] has already been tested on a few classification applications. Specifically, we used GEM score to filter anomalous triplets. GEM score is also beneficial to obtain memory efficient gradients, details will be explained in Section 3.4. Given a derived feature from a knowledge triple $j \in M$, its GEM score $s(l_j)$ is defined using an energy function as,

$$s(l_j) = \log \sum_{k=1}^{K} \exp\left(-\frac{1}{2}(l_j - \mu_k^*)^T \sigma_k^{-1}(l_j - \mu_k^*)\right) \tag{3}$$

where $l_j$ is the textual feature of retrieved external triplets, $\mu_k^*$ is the optimal means and $\sigma_k^*$ is the optimal covariance matrix of ID features.

**Do we require fixed point iterations in Equation (2) to be exact?** In our training pipeline, we only require that the gradients provided by the proposed OOD detection layer to be a descent direction with respect to the loss as a function of network parameters. Thus, the only use of $\mu^*$ and $\sigma^*$ is to calculate OOD score $s$ in Equation (3), so the approximate $\mu^*, \sigma^*$ may be sufficient enough for the training purpose. To see this, we considered a simplified setup in which language features $l$ in Equation (3) are first processed by ReLU based upstream layer parameterized by $W_{\text{up}}$, followed by energy calculation using the output means $\mu_k^*$ of EM algorithm.

Formally, we considered the loss function given by $E(W_{\text{up}}; \mu^*) := -\log(\exp(0.5 \cdot \| \text{relu}(W_{\text{up}} \cdot l) - \mu^*\|_2^2))$ where we assumed number of components to be one i.e., $K = 1$ for simplicity. We now look at the gradients computed by using the approximate output of EM algorithm. Now, the gradients of the energy score (Equation (3)) itself is given by Chain rule as follows:

$$\nabla_{W_{\text{up}}} E(W_{\text{up}}; \mu^*) = \frac{\partial(-\log(\exp(0.5 \cdot \| \text{relu}(W_{\text{up}} \cdot l) - \mu^*\|_2^2)))}{\partial W_{\text{up}}}$$
$$= -(\text{relu}(t_0) - \mu^*) \odot \text{relu}(\text{sign}(t_0)) \cdot l^T \tag{4}$$

where $t_0 := W_{\text{up}} \cdot l$, and $\odot$ denotes the Hadamard or Elementwise product. Importantly, $\nabla_{W_{\text{up}}} E(W_{\text{up}}; \mu^*)$ is linear in $\mu^*$. By definition of a *descent* direction, it is possible to reduce the loss using an approximate $\tilde{\mu}$ computed with finite iterations as long as $\text{tr}\left(\nabla_{W_{\text{up}}} E(W_{\text{up}}; \mu^*)^\top \nabla_{W_{\text{up}}} E(W_{\text{up}}; \tilde{\mu})\right) < 0$. Our calculation above considered only one upstream layer which is rarely the case in practice. We leave extensions to more upstream layers as future work.

---

**Algorithm 1** Fixed Point Network Operator based OOD Detection Layer for Language Features $l_j$

---

**Input**
$\quad x_i \forall i \in [N], l_j \forall j \in [M], \mu_k^0 \forall k \in [K], \sigma_k^0 \forall k \in [K], T$
**Output**
$\quad s(l_j), \nabla_{l_i} s(l_j)(l_j, \mu_i^*, \sigma_i^*)$ $\qquad\qquad\qquad$ ▷ Feature-wise OOD scores and Gradients
*– Begin Forward Pass –*
**while** $t \leq T$ **do** Update $\mu_k^{t+1}$ using Equation (2)
**end while**
Set $\mu_k^*$ to be the last iterate of $\mu_k^T$
**OOD Detection.** Compute $s(l_j)$ in Equation (3)
*– Begin Backward Pass –*
**Jacobian Free Backpropagation.** Output gradient $\nabla_{l_j} s(l_j)(l_j, \mu_i^*, \sigma_i^*)$ by computing derivative of composition of log-sum-exp and ReLU functions in Equations (3) using Chain rule.

---

## 3.3 Implementing Differentiable OOD layer in Multimodal Pipeline

To implement our implicit OOD detection layer discussed in previous sections in multimodal pipelines, we propose a novel architecture called **VK-OOD** by fusing **V**ision and external **K**nowledge features. In essence, we used an **OOD** detector as the network operator $\mathcal{N}$ and detected the retrieved knowledge triplets which can potentially lead to slow convergence of upstream and/or downstream layers during training.

**Architecture.** Given an image with a caption (question), our pipeline consists of the following steps: **(1)** Transform the image to visual features using the vision encoder, such as ViT-based [15], **(2)** Retrieve knowledge triplets using external knowledge bases, and transform to textual features with the language encoder, such as BERT-based [14], more details of retrieval module are introduced in Appendix A, **(3)** Approximate density of in-distribution features, and filter outliers in the image-triplet pairs with the proposed implicit OOD detection layer, **(4)** Finally, we learn vision and textual representations by a multimodal encoder with multiple training objectives.

**Image-text Matching Loss with GEM scores.** We aimed to incorporate the calculated GEM scores to the widely used image-text matching (ITM) loss. Specifically, we consider the model is encouraged to not only match images and texts correctly, but also to map OOD pairs farther away from ID pairs in the feature space. Given image-text pairs $D = \{(v_j, l_j)\}_{j=1}^M$, the similarity of image-text pair based on $s(\cdot)$ is derived as:

$$p(v_q, l_j) = \frac{\exp(m(v_q, l_j)/\tau + \beta \cdot \text{sign}(m(v_q, l_j)) \cdot s(l_j))}{\sum_{u=1}^M \exp(m(v_q, l_u)\tau + \beta \cdot \text{sign}(m(v_q, l_j)) \cdot s(l_u))} \tag{5}$$

where $m(\cdot)$ is the cosine similarity of image-text pair, $\beta$ is a learnable weight parameter, $\text{sign}(\cdot)$ is a sign function. Then, the modified ITM loss is written as:

$$\mathcal{L}_{\text{itm}} = \mathbb{E}_{(v,l) \sim D} \mathcal{H}(y_{itm}, p(v, l)) \tag{6}$$

where $\mathcal{H}$ denotes the cross-entropy, $y_{itm}$ corresponds to the image-text matching label. Note that, $s(\cdot)$ of the same triplets may vary on random masks and different tasks. Hence, we also modified other training objectives losses based on $s(\cdot)$, which are similar to $\mathcal{L}_{\text{itm}}$. See more details in B.

## 3.4 Efficient Backpropagation for OOD Detection Layer

Having found $\mu_k^*, \sigma_k^*$ in the forward pass, we can view the initial part of our OOD network operator $\mathcal{N}$ as the EM algorithm. It outputs the optimal parameters of ID feature density given by $\mu, \sigma$. However, EM viewed as an operator from $\mathbb{R}^d \to \mathbb{R}^d$ (mapping $l_j \in \mathbb{R}^d$ to $\mu_k^* \in \mathbb{R}^d$) makes backpropagation tricky since the Jacobian of such a map will be a $\mathbb{R}^{d \times d}$ matrix, practically infeasible for training purposes even when $d \approx 100$ is not very large. Hence, in this section, we describe how we adopted Jacoian-Free backpropagation for the implicit OOD detection layer.

**Applying Chain Rule with Limited Unrolling for Gradients.** Since each iteration in GMM as in Equation (2) is differentiable, we can easily backpropagate through few iterations of EM algorithm

to compute gradients. First we used Chain rule to illustrate the necessary components to obtain gradients. Note that from Equation (3), we know that $\nabla_s \mathcal{L} \in \mathbb{R}$ is a scalar, and moreover for a fixed $\mu$, $s$ itself is a smooth scalar function $s_\mu : \mathbb{R}^d \to \mathbb{R}$. Thus, we consider the following composition of maps to compute $\nabla_x \mathcal{L} \in \mathbb{R}^d$ gradient with respect to $x_i$ (denoted generically as $x$),

$$x \longmapsto \mu^* \quad \longmapsto s \mapsto \mathcal{L}$$
$$\mathbb{R}^d \xrightarrow{J_{x,\mu^*}} \mathbb{R}^d \longrightarrow \mathbb{R} \to \mathbb{R}, \tag{7}$$

where $\mu^*$ is the output of our OOD layer from forward pass. The key difficulty is in approximating the Jacobian $J_{x,\mu^*} \in \mathbb{R}^d$ and/or its inverse for backpropagation since it maybe dense. We now describe three standard ways that can be used to backpropagate through implicit layers. **First**, in the vanilla backpropagation often called as "unrolling" is used where each iterate is stored, and a "path" gradient approximation is calculated. That is, since the parmeters of each unrolled layer are fixed, we can simply use them along with the closed form gradients available for Equation (2) for backpropagation. **Second,** Jacobian based methods form an approximation to the Jacobian inverse or in other words, solve a linear system formed using the fixed point condition in Equation (2) itself. Often, this approach yields better gradients. **Third**, Jacobian-Free Backpropagation (JFB) combines both these approaches, that is, in which we simply use or unroll the last few iterates of the EM algorithm for gradients. As we can see, JFB enables to train with fixed memory costs ($\mathcal{O}(1)$) and efficient backpropagation without computing gradients at each iterations. The time complexity is $\mathcal{O}(n)$ for forward and $\mathcal{O}(1)$ for backprop. We provide empirical results of the above three methods in Section 4.1.

**Benefits of EM algorithm.** EM updates are provably convergent in various settings, and it takes few iterations when it is guaranteed to converge [13]. This also implies that we can simply initialize $\mu_k$ randomly and perform few more iterations to find a fixed point. However, since the denominator in Equation (2) contains terms that are also unknown parameters, such update schemes may be numerically unstable. In such cases, we can simply use recently proposed gradient based EM algorithms as the network operator, for fine-tuning deep networks in large scale settings, see [51] for convergence analysis.

**Interaction via Outlier Detection.** The applications which require a high number of image patches (or concepts), the likelihood that one of the patch features or text features to be an outlier also increases dramatically. In high dimensional settings, this increases the training time taken by first order methods significantly, especially when mini-batches are used to compute gradients [20]. Alternatively, when features $l_j$ are computationally easy to extract, say using CLIP [49], it is reasonable to expect that a certain fraction of the $l_j$ are outliers, and should not be used for backpropagation purposes. In a more optimistic scenario, we consider to customize our predictions, and handle "on-the-fly" integration of explicit knowledge. In our framework, this corresponds to treating $\mu_k$ in Equation (3) as trainable parameters. We can update the initialization $\mu_k$ without storing the trajectory, or forming the full Jacobian which can be expensive, as in our Algorithm 1.

## 4 Experiments

In this section, we present empirical results addressing the following questions: (1) How effective and robust is the OOD detection layer? (2) What improvement does the Jacobian-Free backpropagation (JFB) provide in computational cost within our VK-OOD architecture? (3) How do external knowledge resources influence the performance of multimodal pipelines? (4) Can VK-OOD work with different backbones in large-scale settings for various downstream tasks? To answer (1) and (2), we performed experiments on VQA task with various setups, comparing memory and training costs with unrolled EM (vanilla), Jacobian-based-EM, and JFB-EM. To answer (3), we evaluated the influences of knowledge resources by explicitly (or intentionally) introducing outliers in the pipeline. Finally, for (4) we utilized ViLT [27], CLIP [49], and BLIP [31] as the backbone model and illustrate performance on downstream tasks, including natural language for visual reasoning and image-text retrieval. More implementation and training details are described in the Appendix C.1. For ablation studies, we fixed the backbone model to be ViLT with pretrained parameters.

### 4.1 Results and analysis of OOD detection layer

**Backprop methods.** We used different backpropgation methods in OOD detection layer with ViLT as the backbone. As shown in Table 1, compared to vanilla and Jacobian-based methods, JFB-EM

| Method | #Param(M) | #FLOPs(G) | Time(m)/epoch Forward | Time(m)/epoch Backward | Max Mem(Mb) | VQAv2 |
|---|---|---|---|---|---|---|
| Vanilla-EM | 152.6 | 185.2 | 39.6 | 26.8 | 18673 | 76.6 |
| JB-EM | 125.2 | 115.7 | 39.6 | 12.7 | 14512 | 76.6 |
| JFB-EM | 124.8 | 108.6 | 39.6 | 6.3 | 13674 | 76.8 |

Table 1: Experimental results of different backpropagation method in the dense OOD detection layer. JFB-EM is much more efficient in backward pass and use less memory. It also outperforms on the VQAv2 task in terms of accuracy.

| k=1 | k=2 | k=3 | k=4 | k=5 |
|---|---|---|---|---|

Figure 2: Visualization of the multimodal feature space. Here, $k$ denotes the number of clusters.

outperformed them in term of accuracy on VQAv2 task. It is trained with significantly less memory and time cost per epoch, which is beneficial for large-scale settings. Additionally, we conducted experiments on $T$ - number of iterations, which is a critical parameter to compute fixed points. The results are shown in Figure 3a. We found the improvements to be marginal for $T \geq 5$, and JFB achieves faster performance on all settings. Based on the results in Figure 3a, we fixed $T = 10$ for the subsequent experimental settings. Additionally, we have provided the fixed point error plot over iterations in Figure 3b. We observed that the squared euclidean distance between successive iterates indeed went to zero showing convergence of forward pass.

**Number of Clusters.** We investigated the effects of the number of clusters on optimizing the GMM process, and the results are presented in Figure 3c (see green line). The general trend indicates that performance improves with an increase in the number of clusters. Qualitatively, Figure 2 shows that learned output features using U-MAP embeddings [42]. Feature embedding spaces of multiple modalities on COCO val dataset are shown in different colors representing different clusters. We provide example images of the clusters in Appendix C.3. The results show that our VK-OOD model can identify clusters over the extracted multimodal features – our layer can accurately detect outliers.

**Robustness of OOD layer.** Usually OOD features are not present within the training datasets themselves. However, we may encounter outliers when integrating external knowledge triplets into the training pipeline. If we denote $M$ as the number of external knowledge triplets, then $M = 0$ corresponds to the ID setup – no outliers. To delve into further quantitative analysis, we considered two setups: one with $M = 0$, i.e., ID setup, and OOD setup with $M = 5$ (so possibly 5 outliers per language caption) that corresponds to augmenting features from external knowledge. We can see from the results in Table 2b that there is not a significant difference from the the rate of convergence perspective — as indicated by squared norm of successive iterates $\|\mu_t - \mu_{t+1}\|_2^2$ — in both setups. However, from the accuracy (Acc) column in Table 2b, we can conclude that the performance in VQA tasks has significant improvements over iterations when considering external knowledge.

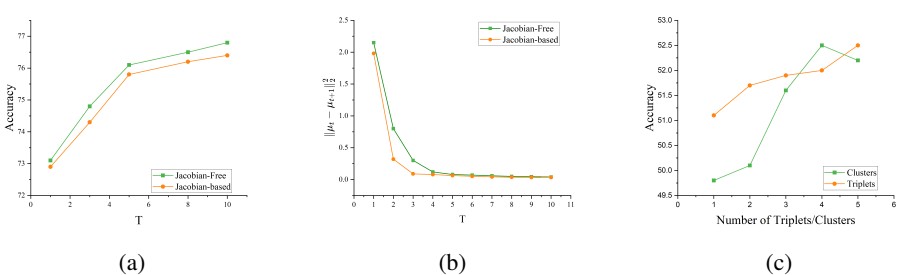

(a)  (b)  (c)

Figure 3: Ablation studies results on VQA task. (a) Results on accuracy with different $T$ iterations in EM on VQAv2 dataset. (b) $\|\mu_t - \mu_{t+1}\|_2^2$ over EM iterations in forward pass.(c) Results on accuracy with different numbers of external triplets and the number of clusters in the GMM process (see Equation (2)) on OKVQA dataset.

**Incomplete Knowledge Triplets.** To evaluate the sensitivity of models to OOD detection performance, we conducted experiments of incomplete knowledge triplets with missing values. Note that, we only dropped language features inputs due to computational reasons – augmenting image patches requires more resources such as GPUs. However, since $x$ is used for both visual and language features, the implementation remains the same as that of dropping patches in language features. So, it is equivalent to drop either the visual or the language feature since the computational effort involved in running EM algorithm depends only on the total number of features. We now present more results of $\|\mu_t - \mu_{t+1}\|_2^2$ and OKVQA performance in term of accuracy over optimization iterations in Table 2a. With higher level of incompleteness, the rate of convergence is slower as expected. Once again, as in previous robustness experiments, we found accuracy gain over iterations here also.

| | 25% | | 50% | | 75% | | | ID(M=0) | | OOD(M=5) | |
|---|---|---|---|---|---|---|---|---|---|---|---|
| T | Err | Acc | Err | Acc | Err | Acc | T | Err | Acc | Err | Acc |
| 1 | 2.213 | 48.2 | 2.448 | 47.3 | 2.735 | 44.6 | 1 | 1.94 | 73.6 | 2.15 | 73.1 |
| 3 | 0.174 | 48.9 | 0.214 | 47.7 | 0.208 | 45.1 | 3 | 0.059 | 73.8 | 0.089 | 74.8 |
| 5 | 0.065 | 50.6 | 0.092 | 48.6 | 0.244 | 46.0 | 5 | 0.038 | 73.9 | 0.051 | 76.1 |
| 8 | 0.057 | 51.2 | 0.108 | 49.1 | 0.112 | 46.6 | 8 | 0.042 | 73.9 | 0.036 | 76.5 |
| 10 | 0.051 | 51.8 | 0.084 | 49.4 | 0.167 | 46.9 | 10 | 0.036 | 74.1 | 0.034 | 76.8 |

(a) $\mu$ error and OKVQA accuracy        (b) $\mu$ error and VQA accuracy

Table 2: Fixed point error is measured using consecutive iterate distance $\|\mu_t - \mu_{t+1}\|_2^2$. (a) $\mu$ error and OKVQA accuracy over the EM optimization iterations in different level of incompleteness. (b) $\mu$ error and VQA accuracy over the EM optimization iterations in ID and OOD setups.

## 4.2 Ablation study on VK-OOD Components and External Knowledge

**Effectiveness of Each Component.** To compare the impact of the proposed OOD layer in VK-OOD, we considered different combinations of inclusion and exclusion of knowledge graph representations (KG) and OOD detection layer. The results are shown in Table 3. The results show that our model achieves the best performance when both the components are included in the model. Moreover, comparing the results on VQAV2 and OKVQA datasets, the results imply that external knowledge triplets (KG) can be beneficial to improve the performance especially on VQA task. Furthermore, using OOD layer even when there are no external knowledge triplets has good performance. This shows that including OOD layer in our model is helpful and able to capture the noise of multiple modalities, such as missing or mismatching modalities.

**Numbers of external knowledge triplets.** We conducted experiments to explore the impact of the amount of the knowledge triplets. We evaluate this on VQA tasks using OKVQA dataset. Figure 3c shows the experimental results. As expected, we observed that increasing the number of retrieved knowledge triplets improve the accuracy of predicted answers. We achieved the best accuracy of $52.4\%$ when the number of triplets is $\approx 5$.

| Method | | Downstream tasks | |
|---|---|---|---|
| KG | OOD | VQAV2 | OKVQA |
| | | 73.9 | 45.5 |
| ✓ | | 74.6 | 48.3 |
| | ✓ | 74.1 | 46.2 |
| ✓ | ✓ | **76.8** | **52.4** |

Table 3: Ablation studies of different components of VK-OOD. "KG" and "OOD" denote knowledge graph and OOD detection layer respectively.

**Knowledge resources.** We evaluated different knowledge resources, i.e., different embeddings of implicit and/or explicit knowledge. Table 4 shows benefits provided by different external resources in our pipeline. We queried knowledge from Wikidata [59] and used BERT to get embeddings $l_j \in \mathbb{R}^d$. Our model produced 1.8% and 18.7% more accurate results than the best and worst performing baselines respectively. Moreover, using ConcepNet embeddings solely, our multimodal training pipeline also learn implicit knowledge in the multimodal fusion encoder. We compared our model performance with the models using external knowledge resources. As results shown in Table 4, all the different external knowledge resources brought improvements in the OKVQA tasks. Our proposed model takes advantages of implicit knowledge from vision-language models and integrates explicit knowledge prior information, thus outperforms other models using external knowledge resources without OOD detection.

## 4.3 Scalability of VK-OOD

In this section, we incorporated the proposed OOD detection layer to various backbone models with different architectures and model sizes. We initialized the parameters ($\mu$ and $\sigma$) of our proposed

OOD detection layer in two different ways, scalar $\sigma$ or the dense one which $\sigma$ is a $d \times d$ matrix, where d is the dimension of input embeddings. As showin in Table 5, the number of parameters in scalar VK-OOD are approximately similar to the baselines. Specifically, comparing to other baseline models, while our *scalar* VK-OOD increases the #-parameters slightly – $\approx 0.4$ million more parameters (since $d \approx 700$) – it **significantly** improved the performance in downstream tasks.

| Method | Knolwedge resources | OKVQA |
|---|---|---|
| ConceptBERT | CN | 33.7 |
| KRISP | Wiki + CN | 38.4 |
| MAVEx | Wiki + CN + GI | 39.4 |
| KAT-B | Wiki + GPT3 | 50.6 |
| UnifER | CN + ViLT | 42.1 |
| | CN - w/o BERT | 51.1 |
| VK-OOD (Ours) | Wiki | 51.9 |
| | Wiki + CN | 52.2 |
| | CN | **52.4** |

Table 4: Results on the different knowledge resources on OKVQA dataset. "CN" and "GI" denote ConceptNet and Google Images respectively.

We then conducted experiments on several downstream tasks. In all these experiments, our model consistently achieved the best and second-best performance compared to *five state-of-the-art (SOTA)* vision-language models across *three downstream tasks* on various datasets(see Table 5). We trained on open-source data and compare the results with the models having similar number of parameters. In particular, for VQA tasks, we evaluated on the VQAv2 test set. As shown in Table 5, our model outperformed all the baselines on this dataset, yielding an accuracy of 77.9%.

Furthermore, in the Natural Language for Visual Reasoning (NLVR) task, VK-OOD achieved the best and second-best result in terms of accuracy with different backbone models, respectively. We also evaluated our model, along with the baseline models, on the test set of COCO and F30K dataset for image-text retrieval task. Our model produced the best performance and outperformed the best and worst-performing baselines, with the exception of BLIP, by up to 0.5% and 11.6% respectively on the COCO dataset (Table 5). In other settings and on the F30K dataset, the results are similar. Further details are provided in Appendix C.2.

The performance of our model demonstrates its ability for visual reasoning while integrating both implicit and explicit knowledge. Moreover, we are able to show that the proposed OOD detection layer can scale with different models and contribute improvements to various tasks. For inference, since $\mu$ and $\sigma$ are fixed, the inference time is similar to the backbone models considered – the average inference time for one VQA instance is around 57 ms on a single 2080Ti GPU. Therefore, we believe that our implicit layer can function as a plug-and-play module and can be easily integrated with other vision-language models.

### 4.4 Qualitative Analysis

**Visualizing Attention with Multimodal Information.** Figure 4 is an example of multimodal alignment results from our VK-OOD representations. We use Grad-cam [52] to visualize the multimodal maps of the models on the image corresponding to knowledge triplets. As shown in

| Model | #Params | VQAv2 | NLVR2 | COCO | | Flickr30k | |
|---|---|---|---|---|---|---|---|
| | | | | TR R@5 | IR R@5 | TR R@5 | IR R@5 |
| ViLT | 87 | 70.3 | 74.6 | 86.2 | 72 | 95.6 | 86.8 |
| UNITER | 155 | 72.7 | 75.8 | 87.4 | 78.5 | 97.1 | 92.4 |
| ALBEF | 314 | 74.5 | 80.5 | 91.4 | 81.5 | 99.4 | 96.7 |
| VinVL | 157 | 75.9 | 83.1 | 92.6 | 83.2 | - | - |
| BLIP* | 346 | 77.5 | 82.8 | 95.2 | **85.4** | **99.8** | **97.5** |
| VK-OOD-s(ViLT) | 87.4 | 76.7 | 84.3 | 90.9 | 81.6 | 97 | 94.3 |
| VK-OOD-s(CLIP) | 113.4 | 76.2 | 83.8 | 92.8 | 83.4 | 99.6 | 96.7 |
| VK-OOD-s(BLIP) | 346.4 | 77.8 | 84.1 | **95.4** | 85.2 | **99.8** | 97.2 |
| VK-OOD-l(ViLT) | 125 | 76.8 | **84.6** | 91.7 | 81.3 | 97.2 | 94.5 |
| VK-OOD-l(CLIP) | 151 | 76.1 | 83.9 | 93.1 | 83.6 | 99.6 | 96.8 |
| VK-OOD-l(BLIP) | 412 | **77.9** | 84.5 | 95.1 | 84.8 | 99.6 | 97.1 |

Table 5: Overall performance on multiple downstream tasks. We demonstrate VK-OOD scale with different model backbones and achieve the best and second-best results. VK-OOD-s is the scalar case, and VK-OOD-l is the dense case. *our implementation.

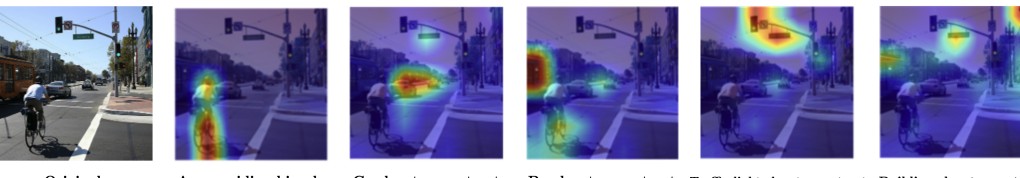

| Original | A man riding bicycle | Car locates on street | Bus locates on street | Traffic lights locate on street | Buildings locate on street |

Figure 4: Visualization of the attention maps of image and triplets alignment. The original sample caption is "A man riding a bicycle down a city street". We highlight areas in the image corresponding to different knowledge triplets. Our model learns different objects and localize those objects correctly.

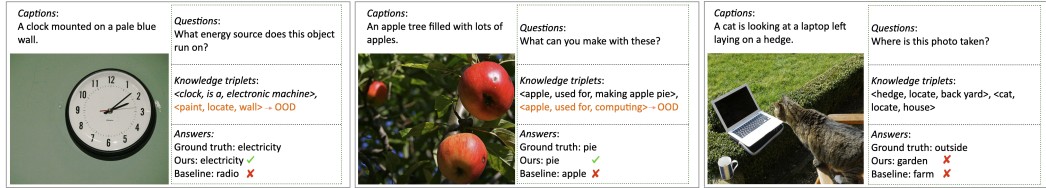

Figure 5: Example case studies with OKVQA dataset. We show the examples of retrieved knowledge triplets. Our model is able to detect outliers of retrieved triplets shown in orange. The predicted answers are finetuning on OKVQA dataset. Comparing with the baseline results, our model provides more correct answers. Note that, the baseline model here is our implementation without KG and OOD components.

Figure 4, our model has the capability to attend to the extracted knowledge concepts, such as buildings and traffic lights. Thus, our model can detect more objects to provide the ability for answering open questions. We discuss more user studies on interactive OOD detections by feeding in domain knowledge with different distributions in the Appendix C.3.

Furthermore, we present prediction results in Figure 5 with VK-OOD model on OKVQA dataset along with the extracted knowledge triplets based on the captions. In the example, the caption ⟨apple, used for, making apple pie⟩ is useful to obtain correct answers. This observation validated that explicit knowledge provides more reasoning capability than implicit knowledge. Moreover, our model detected OOD triplets by combining modalities, i.e, the apple fruit is in image, thus is not used for computing. The last one is a failure case, because the ground truth caption is not sufficiently specific. Therefore, it might be beneficial to consider more inference and reasoning abilities in multimodal analysis, such as chain of thoughts [62].

## 5 Discussion

**Limitations.** A potential limitation of our proposed implicit layer arises when the covariance matrix $\sigma$ is dense. In such cases, a fast linear system solver would be required to evaluate the likelihood. We will consider to explore sparse approaches to further saccelerater OOD detection layer in the future. Additionally, given that we have demonstrated that explicit knowledge can serve as supervision in vision-language training, we believe that various knowledge bases, such as medical knowledge graphs, can provide user-desired domain distributions.

**Conclusions.** In this work, we presented a training framework designed to facilitate multimodal analysis under distribution shifts and/or the presence of outlier distributions within the input feature space. Several other models have been proposed to exploit special structures in the available modalities for faster training purposes, as mentioned in [61], and for tasks involving egocentric vision [64]. While the approaches have been shown to perform well in large-scale settings, they may not be sufficient on their own. For instance, most frames in a video have low semantic information content and may require complex processing pipelines [8]. We argued that handling outliers within the context of multimodal analysis is an crucial topic as more models are integrated or fused. Our implicit OOD detection layer can be directly instantiated within such complex pipelines, possibly allowing us to intervene and accelerate the training process while reducing computation costs. We demonstrated through extensive empirical analysis across various setups that incorporating OOD detection in the training pipeline can significantly enhance the performance on downstream tasks.

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

# A More details of Multimodal Pipeline

## A.1 Image Encoder

We extracted features from image patches and used vision transformer based models(ViTs) [58] as our encoder for the patches. To do so, we split input image into a sequence of patches and project them using linear maps to obtain embedding of patch features $v_q$, which simplifies the step for fusing with text embedding. Following the vision-language models, we trained our model with multiple popular ViTs to examine the influence of image encoder in OOD detection backpropagation process.

## A.2 Knowledge Retrieval Module

Given the caption $S$, we first parsed it into triplets in the form of $T^{id} = \langle o(c), r(c), o'(c) \rangle$, where $o(c)$ and $o'(c)$ are concepts $\in C^{id}$, and $r(c)$ is the relation(s) between them, i.e., $\langle$man, riding, bicycle$\rangle$. In our example, the seed triplets (ID triplets) parsed from the caption are $\langle$man, riding, bicycle$\rangle$ and $\langle$bicycle, down, street$\rangle$. Then we augmented knowledge by matching these triplets with external open knowledge including domain and commonsense knowledge graphs, i.e., ConceptNet [55]. ConceptNet provides a large scale commonsense knowledge with over 21 million edges by 36 type of relations, i.e., IsA, UsedFor, AtLocation, connecting 8 million nodes. To complete our knowledge graph based augmentation, we collected concepts by querying from ConceptNet using $o(c)$, $o'(c)$ and $rel_i$ where $i \in [0, 36]$ and concatenated them to seed triplets. For example, given "street" as $o(c)$ and "AtLocation" as $rel_i$, we will extract the related concepts are located at street to form triple $T_i$. Specifically, we queried explicit knowledge triplets of $o(c)$ and $o'(c)$ from ConceptNet to form $T^{cn}$, i.e., $\langle$bicycle, used for, transport$\rangle$. Finally, these knowledge triplets $\in T = T^{cn} \cup T^{id}$ are encoded as language features, such as $l_j$ using a language encoder (i.e., BERT[14]).

## A.3 Multimodal Fusion Encoder

We used our proposed OOD detection layer as a "plug-and-play" module that can be used in different vision-language architectures. In ViLT and CLIP, we added the visual and textual embeddings with $s(\cdot)$ as in Equation (3), and passed them to a standard L-depth transformer. While using the BLIP backbones, we implemented a two-stream transformer pipeline consisting of stacked multiple layers to joint vision and knowledge textual representations. For each layer, we used self-attention unit and merged cross-attention unit, thus integrating vision and knowledge semantic information and the alignments across them.

**Multimodal Attention Module.** As in the standard transformer architecture [58], the attention function computes identical learnable parameters (weights) as in Equation (8) and Equation (9), where d is the dimension of the inputs, a query Q, key K, and value V. We used fusion encoder to compare similarity among the image-text pairs given by,

$$\text{Attn}\,(Q_I, K_L, V_L) = \\ \text{softmax}\left(\frac{Q_I K_L^T}{\sqrt{d}}\right))V_L, \tag{8}$$

and

$$\text{Attn}\,(Q_L, K_I, V_I) = \\ \text{softmax}\left(\frac{Q_L K_I^T}{\sqrt{d}}\right))V_I \tag{9}$$

where $I$ and $L$ correspond to image modality and language modality.

In implementation of multimodal encoder, we updated the image and language embedding outputs themselves from previous layer as queries and concatenate them together to get keys and values. As a further improvement of attention function, we used a multi-head attention which is composed by multiple paralleled attention function in each head. The feed-forward layers transform the outputs of multi-head attention through two fully-connected layers with GeLU activation.

# B More details on formulating Training objectives

We provide more details on our training objectives in our pipeline in this section, including image text matching (ITM) and masked language modeling (MLM).

## B.1 Image Text Matching (ITM)

To incorporate both the vision and the language representations, we used ITM which is widely used in previous VL studies. Given an image and text of triple pair $\langle v_q, l_j \rangle$, ITM predicts whether they are matched as positive examples or not, and it is a binary classification problem with the loss function in Equation (5) and Equation (6). We assumed that each image and ID triple pair $\langle v_q, l_j \rangle$, as a positive example. The negative pairs are constructed through batch-sampling.

## B.2 Masked Language Modeling (MLM)

MLM utilizes vision features and text features of ID concepts and relations to predict the masked tokens in the caption sentence $S$. Masking tokens is a form of self-supervision, and is well known to improve performance [34]. Here, we randomly masked some tokens in $S$ replacing as $y^{msk}$ and predicted them with their visual and textual features. Since some tokens are replaced with "[mask]", the OOD score $s(\cdot)$ becomes a function of the random masks. With this, we used $s(\cdot)$ to calculate the predicted probability for a masked token as in Equation (3). Our final MLM loss can be written as,

$$\mathcal{L}_{\mathrm{mlm}} = \mathbb{E}_{(v,\hat{l}) \sim D} \mathcal{H}(y_{mlm}, p(v, \hat{l})) \tag{10}$$

where $\mathcal{H}$ denotes the cross-entropy, $y_{mlm}$ is a one-hot vector where the ground truth tokens are with probabilities of $1$, $\hat{l}$ denotes the masked text.

# C Additional Experiments

In this section, we provide details on experimental setups that we used to conduct ablation studies, and more qualitative analysis of our proposed VK-OOD multimodal pipeline.

## C.1 Implementation details

**Datasets.** Following standard practice in Vision, we used training strategies wherein we pre-trained on a fixed, large dataset and then fine-tuned on datasets specific to downstream tasks. We pre-trained on three datasets, including COCO [35], Visual Genome [28], and SBU Captions [47] with total of 1M images and 6.8M image-caption pairs, as approximate $30\%$ less than the baseline(ViLT). Each caption is parsed to 1 - 3 triplets and augmented with 5 external knowledge triplets. For downstream datasts, we used Flickr30k [48] and COCO for image-text retrieval, VQAv2 [4] and OKVQA [41] for visual question answering (and ablation studies), and NLVR2 [56] for visual reasoning. We resized each image to the size of $224 \times 224$ by center-cropping. In the merged attention module, each multimodal encoder layer consists of one multi-head self-attention block and one feedforward block, and total number of identical layers is equal to 12.

**Encoder backbones.** First, we retrieved explicit knowledge triplets as pre-processing, by using ConceptNet Numberbatch[1]. Next, for ViLT, we used RoBERTa [38] as text encoder and ViT-B/32 [49] as visual encoder. To scale with CLIP, we used CLIP-ViT-B/32 [49] as both backbones. Then, we followed the BLIP design with BERT-base as text encoder and ViT-B/16 as visual encoder.

**Network training.** We pre-trained the model for 10 epochs using AdamW optimizer [39] with learning rate of $1e-4$ and weight decay of $1e-2$. We chose the warm-up phase of learning rate to be $10\%$ of the total training steps, and the learning rate was decayed linearly to $0$ afterwards. Then, we fine-tuned our model for 5 epochs with learning rate of $2e-4$ for all downstream tasks. In addition, we applied RandAugment [12] as augmentation strategy in fine-tuning steps. We pre-trained and fine-tuned both on 8 NVIDIA RTX 2080Ti GPUs, and for inference we used 1 NVIDIA RTX 2080Ti GPU.

---

[1] https://github.com/commonsense/conceptnet-numberbatch

| Model | COCO | | | | | | F30k | | | | | |
|---|---|---|---|---|---|---|---|---|---|---|---|---|
| | TR | | | IR | | | TR | | | IR | | |
| | R@1 | R@5 | R@10 | R@1 | R@5 | R@10 | R@1 | R@5 | R@10 | R@1 | R@5 | R@10 |
| ViLT | 61.8 | 86.2 | 92.6 | 41.3 | 72.0 | 82.5 | 81.4 | 95.6 | 97.6 | 61.9 | 86.8 | 92.8 |
| UNITER | 64.4 | 87.4 | 93.1 | 50.3 | 78.5 | 87.2 | 85.9 | 97.1 | 98.8 | 72.5 | 92.4 | 96.1 |
| ALBEF | 73.1 | 91.4 | 96.0 | 56.8 | 81.5 | 89.2 | 94.3 | 99.4 | 99.8 | 82.8 | 96.7 | 98.4 |
| VinVL | 74.6 | 92.6 | 96.3 | 58.1 | 83.2 | 90.1 | - | - | - | - | - | - |
| BLIP | 80.6 | **95.2** | **97.6** | **63.1** | **85.3** | 91.1 | **96.6** | **99.8** | **100** | 87.2 | **97.5** | **98.8** |
| Ours(ViLT) | 73.8 | 91.4 | 96 | 52.4 | 81.3 | 90.1 | 85.9 | 97.1 | 97.6 | 80.1 | 94.6 | 96.7 |
| Ours(CLIP) | 69.8 | 87.5 | 93.6 | 48.8 | 78.5 | 82.5 | 92.3 | 98.4 | 99.5 | 79.8 | 92.1 | 96.4 |
| Ours(BLIP) | **80.7** | 95.1 | 96.8 | 62.9 | 84.8 | **92.8** | 96.4 | 99.6 | 99.8 | 86.3 | 97.1 | **98.8** |

Table 6: Detailed results of image-text retrieval tasks on COCO and Flickr30k datasets. Our model with different backbones outperforms other models and achieve the best and second-best results.

| Model | Objectives | VQA | Flickr30k | |
|---|---|---|---|---|
| | | test-dev | TR@1 | IR@1 |
| ViLT | ITM | 70.6 | 82.1 | 65.6 |
| ViLT | MLM | 72.8 | - | - |
| ViLT | ITM+MLM | 74.2 | 88.1 | 74.1 |
| VK-OOD-s | ITM | 72.1 | 84.5 | 69.8 |
| VK-OOD-s | MLM | 73.4 | - | - |
| VK-OOD-s | ITM+MLM | **74.8** | **89.0** | **77.2** |

Table 7: Ablation study experiment results of VK-OOD model. ViLT is our implementation without explicit knowledge and OOD detection layer. ITM is image-text matching, and MLM is masked language modeling. Results on VQA are on test-dev set. Both downstream results are in zero-shot settings. The bold values mean the best model in the table. Comparing with the baselines, our model with OOD detection layer outperforms on all objectives with two datasets. Training on combinations of objectives improves model performance.

## C.2   Experimental Results

**Details of image-text retrieval tasks.** We now discuss detailed results on COCO and F30K datasets, as shown in Table 6. We can see that the model of OOD detection layer with ViLT has significant improvements in image retrieval and text retrieval tasks. Overall, we once again see that our model achieved the best and second-best results on both datasets comparing to other SOTA models.

**Training Objectives with OOD detection Layer.** To precisely characterize the performance benefits of using our OOD layer, we performed more ablations with the default training settings of the baseline and our model mentioned in Section 4.2. We considered different combinations of train objectives and evaluated them in zero-shot settings and observed our model performance on training objectives. Note that ViLT we compare to in Table 7 is our implementation with the same subset of training datasets. Our raw results are presented in Table 7. Here, we trained on pre-train datasets with $\mathcal{L}_{\text{itm}}$ in Equation (6), $\mathcal{L}_{\text{mlm}}$ in Equation (10), and the sum of $\mathcal{L}_{\text{itm}}$ and $\mathcal{L}_{\text{mlm}}$ losses as our training objectives. We see that training on image-text matching and masked language modeling is beneficial for both downstream tasks comparing to the baseline model. Specifically, there is improvements in image retrieval and text retrieval tasks. Thus, it is beneficial to train on both ITM and MLM for filtering outlier concepts and improved performance on downstream tasks.

## C.3   Qualitative Analysis

As shown in Figure 2, the feature maps obtained using our VK-OOD pipeline can be clustered that are easily identified. This means that we can detect outliers and group images closest to the corresponding $\mu_k$ with image and explicit knowledge triplets. Figure 6a and Figure 6b are examples with the nearest images in each cluster.

Figure 7 and Figure 8 show more qualitative examples of multimodal alignment results of our models. Here, we visualized the multimodal attention maps on images corresponding to concept triplets using Grad-cam [52]. Following our model architecture, the caption is parsed and integrated with knowledge triplets. The right bottom sub-figures of our model in Figure 7 and Figure 8 are the

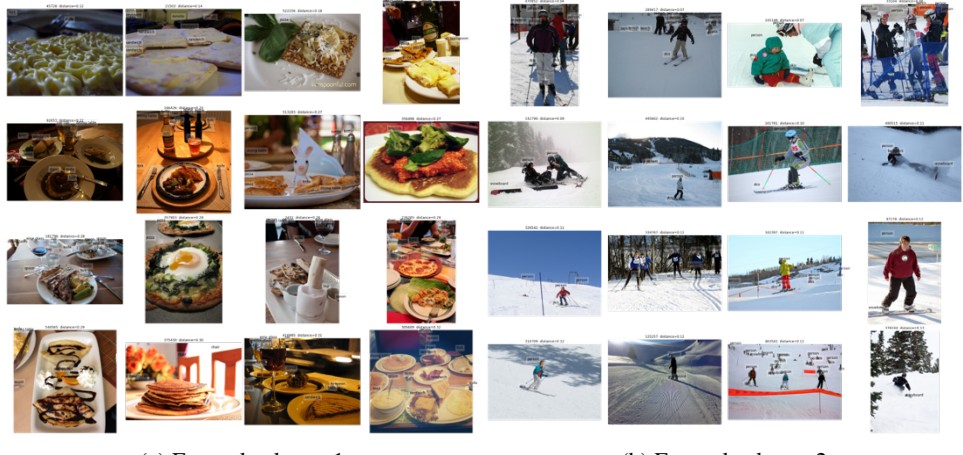

(a) Example cluster 1                    (b) Example cluster 2

Figure 6: Example images in the clusters on COCO val set.

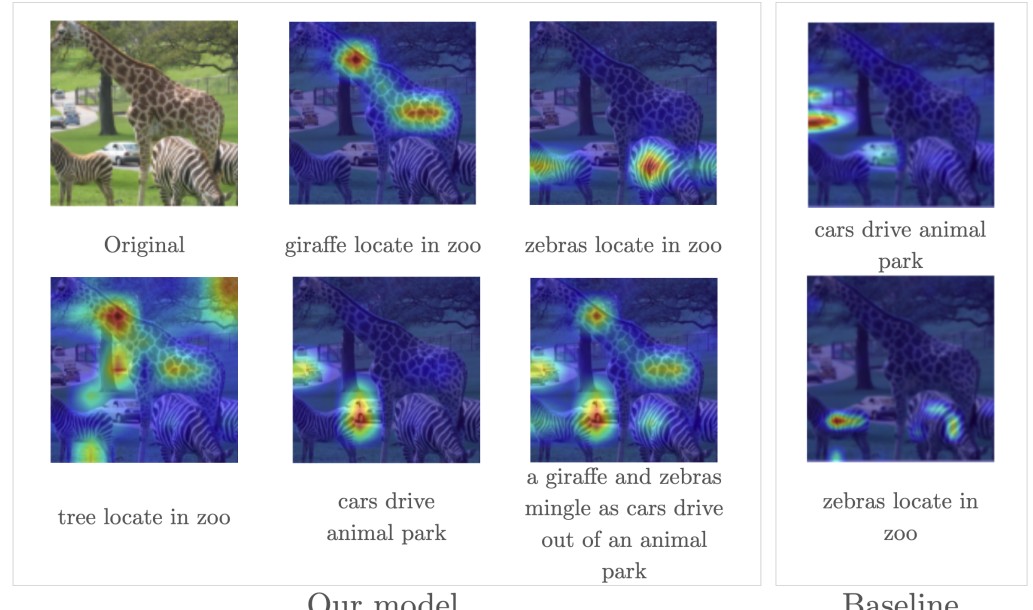

Figure 7: Visualization of the attention maps of image feature $v_q$ and language features $l_j$ of external knowledge alignment. The results are from our VK-OOD-s (ViLT) model. The original sample caption is "a giraffe and zebras mingle as cars drive out of an animal park". We highlight areas in the example image corresponding to different knowledge triplets. Comparing with the attention maps of the baseline model, our model learns object shapes such as zebras and localize those objects correctly.

multimodal alignment of original captions from MSCOCO dataset [35]. Other sub-figures show the alignments of extracted knowledge triplets on the image.

Interestingly, we found that VK-OOD model is able to capture concept "plug" as a part of "refrigerator" or "microwave" in Figure 7. The heatmap area of "plug" and "microwave" in Figure 7 clearly suggest that our model has the capability to exploit different relevance between visual and corresponding conceptual text features. In contrast, the baseline results did not show the relation between plug and microwave. Figure 8, shows that VK-OOD can detect three zebras comparing with baseline, but counting cars is not performing well as we expected – since the size (or scale) of cars is not sufficiently high, and moreover some parts of them are occluded.

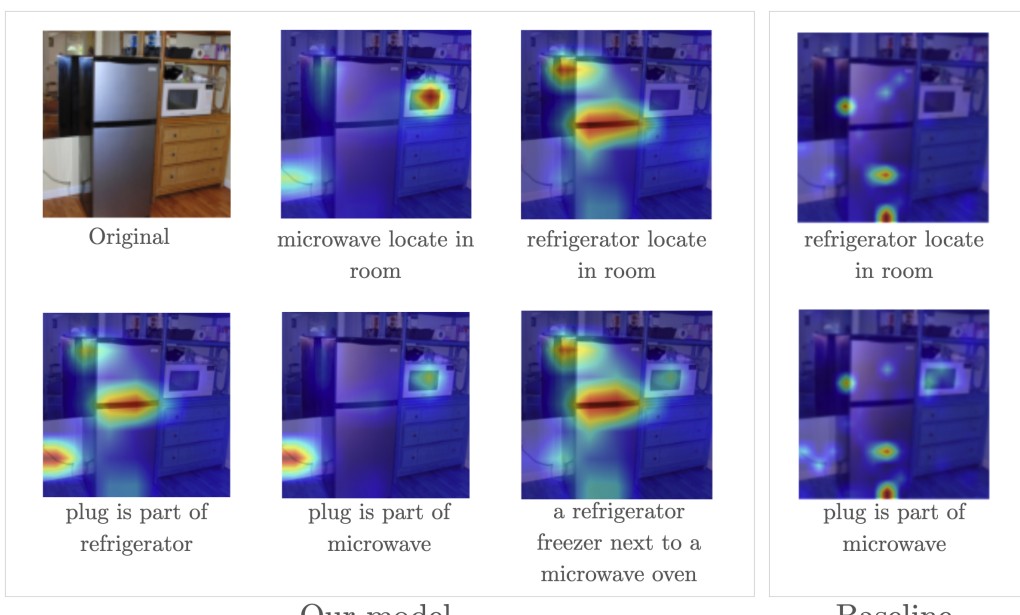

Figure 8: Visualization of the attention maps of image and knowledge concept triplets alignment. The results are from our VK-OOD-s (ViLT) model. The original sample caption is "a metallic refrigerator freezer next to a microwave oven". We highlight areas in the example image corresponding to different knowledge triplets. Comparing with the attention maps of the baseline model, our model learns the relations between the parts (i.e., plug) of the objects correctly.

