# OpenReview forum: "Implicit Differentiable Outlier Detection Enable Robust Deep Multimodal Analysis"
_NeurIPS.cc/2023/Conference — NeurIPS 2023 poster_

### Official Review · Reviewer_NYTC · 2023-06-17

**Soundness:** 3 good
**Presentation:** 4 excellent
**Contribution:** 3 good
**Rating:** 5
**Confidence:** 4

**Summary:**

The paper presents a method for incorporating outlier detection into an end-to-end learnable framework. Essentially the paper shows how it is possible to differentiate efficiently through a per-instance Gaussian mixture model using either unrolling, implicit differentiation or Jacobian-free back-propagation as discussed in recent literature. The paper is well written and experiments combining vision and language for visual-question answering (VQA) are promising.

**Strengths:**

The paper is well written and the proposed method for a differentiable Gaussian mixture model is sound. I am mostly positive about the paper but have a few questions below. I am willing to raise my score depending on the answers provided by the authors.

**Weaknesses:**

As mentioned above, the paper is mostly well written. However, there are a few minor places where the paper could be improved. Some prior related work on differentiable optimization is missing (e.g., Gould et al., TPAMI 2022). These and cited prior related works already discuss the reduced memory costs during back-propagation via implicated differentiation limiting the novelty of the paper (certainly Line 73). Some of the mathematical expressions can be tidied, e.g.,

a. Size of outer parentheses in Eqn (3)
b. Lower limit in denominator summation in Eqn (4) should be "u=1"

And some consistency in referencing tables and figures. E.g., Section 4.1 refers to figures in four different ways "Figure 3a", "Figure3a" (no space), "Fig 3b" (no period), and "Fig. 2" (period, but breaking space). Also tables are referenced with lowercase "table" and figures with uppercase "Figure". I recommend being consistent and using non-breaking spaces between label and reference number.

Also, wrapping tables in the text can result to confusing line breaks, e.g., Line 265.

The above are all minor presentation issues. Two more serious concerns are:

1. The results from implicit differentiation are only valid when the forward pass algorithm actually results in a fixed point, i.e., is run to convergence. Since the method presented in the paper only iterates for a pre-determined number of iterations, convergence cannot be guaranteed. What happens when the resulting GMM parameters have not converged? Can the authors comment on the validity of the gradients obtained?

2. Table 1 compares three different methods for back-propagation. I understand that the time and memory of these methods will vary but I don't understand why the number of parameters in the models should be different (i.e., 152.6M, 125.2M and 124.8M for Vanilla-EM, JB-EM and JFB-EM, respectively). Can the authors please comment. Also, does the time per epoch include execution of the forward pass? I would expect this to dominate the backward pass. Can the authors please provide forward pass timings separate from backward pass timings.

**Questions:**

See questions raised in Weaknesses above. I am willing to increase my score depending on the answers to these questions, especially relating to results reported in Table 1.

**Limitations:**

Limitations were adequately addressed.

---

> ### Author Rebuttal · Authors · 2023-08-10
>
> Thank you for taking the time to review our paper. We appreciate your constructive suggestions to illustrate forward pass and backward pass execution time separately in Table 1. We will fix all the typos including in math expressions and keep the references of tables and figures consistent in the revised version. We will address your concerns in the following responses:
>
> >The results from implicit differentiation are only valid when the forward pass algorithm actually results in a fixed point, i.e., is run to convergence. Since the method presented in the paper only iterates for a pre-determined number of iterations, convergence cannot be guaranteed. What happens when the resulting GMM parameters have not converged? Can the authors comment on the validity of the gradients obtained?
>
> **Concern 1:**
> Please refer to the general response 3 to see if it is answered your question. As long as the optimization algorithm (JFB-EM) converges to clusters points $\tilde{\mu},\tilde{\sigma}$ such that the gradients calculated is positively correlated to the true gradients, we can decrease the loss function. Thus, it is not necessary that EM iterations solve the clustering problem exactly. In  practice, we run for few iterations to get a reasonable $\mu$ and $\sigma$ and use the energy of GEM score to filter OOD samples in the overall pipeline. Thus, the only use of $\mu^*$ and $\sigma^*$ is to calculate OOD score $s$ in equation (4), so the approximate $s$ is sufficient enough for optimization purposes.
>
> >Table 1 compares three different methods for back-propagation. I understand that the time and memory of these methods will vary but I don't understand why the number of parameters in the models should be different (i.e., 152.6M, 125.2M and 124.8M for Vanilla-EM, JB-EM and JFB-EM, respectively). Can the authors please comment. Also, does the time per epoch include execution of the forward pass? I would expect this to dominate the backward pass. Can the authors please provide forward pass timings separate from backward pass timings.
>
> **Concern 2:**
>
> (1) We need to store the intermediate gradients and $\mu$ and $\sigma$ of each iteration for Vanilla-EM and full Jacobian matrix for JB-EM during training, so we consider to add them to the number of parameters.
>
> (2) Yes, the time reported in Table 1 includes the execution time of the forward pass.
>
> (3) We agree with the reviewer that forward pass dominates backward pass since we can reuse the weights available during forward pass.
>
> (4) We update Table 1 with separate forward pass and backward pass timings in the rebuttal PDF and will update it in the revised version.

---

> > ### Comment · Reviewer_NYTC · 2023-08-17
> >
> > Thanks for the detailed response. I accept that running EM for a few iterations may give a good approximation to the gradient (sometimes) and work well in practice. I think you have to be careful about claiming that all you need is positive correlation with the approximate and true gradients, i.e., $g_{\mu*, \sigma*}^T g_{\mu, \sigma} \geq 0$. First, this can't be known without actually computing the true stationary values of $\mu$ and $\sigma$. Second, the chain rule does not hold for approximated gradients so just because there is positive correlation with gradients of the loss with respect to $\mu$ and $\sigma$ does not mean that there will be a positive correlation with the gradients of the loss with respect to upstream parameters. Discussion/warning of this should be included in the paper.

---

> > > ### Author Response · Authors · 2023-08-19
> > >
> > > Dear Reviewer NYTC,
> > >
> > > Thank you for the further discussions. Here, we will clarify what we meant by correlation for a more enthusiastic support of our submission. An upstream layer is any layer before a certain layer -- in our case VK-OOD layer --  and so is closer to input. To simplify terminology, we consider a simplified setup in which language features $l$ in equation (3) are first processed by ReLU based "Upstream" layer parametrized by $W_{\text{up}}$, followed by energy calculation using the output means ${\mu}_k^*$ of EM algorithm. We now look at the gradient computed using an approximate output of EM algorithm.
> > >
> > > Formally, we consider the loss function given by $$E(W_{\text{up}};\mu^*):=-\log (\exp (0.5 \cdot  \|\ \text{ReLu}(W_{\text{up}} \cdot l) - \mu^* \|_2^2 ))$$
> > >
> > > where we assumed number of components to be one i.e., $K=1$ for simplicity. Now, the gradient of the energy score ( equation (2)) itself is given by Chain rule as follows:
> > > $\nabla_{W_{\text{up}}}E(W_{\text{up}};\mu^*) =$ $$ \frac{\partial (-\log (\exp (0.5 \cdot |\ \text{ReLu}(W_{\text{up}} \cdot l) - \mu^* |_2^2 )))} {\partial W{\text{up}}} $$
> > >  $$   = - (\text{ReLu}(t_0) - \mu^*) \odot \text{ReLu}(\text{sign}(t_0)) \cdot l^T
> > > $$
> > >
> > > where $t_0 := W_{\text{up}} \cdot l$, and $\odot$ denotes the Hadamard or Elementwise product. Importantly, please note that $\nabla_{W_{\text{up}}}E(W_{\text{up}};\mu^*)$ is linear in $\mu^*$.  By definition of a {\em descent} direction, it is possible to reduce the loss using an approximate $\tilde{\mu}$ computed with finite iterations as long as $\text{tr}\left(\nabla_{W_{\text{up}}}E(W_{\text{up}};\mu^*)^{\top} \nabla_{W_{\text{up}}}E(W_{\text{up}};\tilde{\mu}) \right)<0 $. In light of this linear relationship, we mentioned the correlation in our response.
> > >
> > > We agree with you that this condition is not known to us without actually computing $\mu^*$, and this relationship heavily depend on ReLU layers. We are happy to clarify this in our revision, and consider extensions in our future work. Kindly inform us if our responses addressed your concerns satisfactorily, and do not hesitate to request further elaboration if needed. Thank you once again for your constructive feedback.

---

> > > > ### Author Response · Authors · 2023-08-21
> > > >
> > > > Dear Reviewer NYTC,
> > > >
> > > > We would greatly appreciate any further feedback on our rebuttal. We hope that we have adequately addressed your previous concerns, and we are more than happy to answer any other remaining questions you may have. We sincerely value your feedback. Many thanks in advance!

---

> > > > ### Comment · Reviewer_NYTC · 2023-08-21
> > > >
> > > > Thank you. It would be valuable to clarify this discussion in the paper. I will raise my score.

---

### Official Review · Reviewer_MwwQ · 2023-07-06

**Soundness:** 3 good
**Presentation:** 3 good
**Contribution:** 2 fair
**Rating:** 5
**Confidence:** 4

**Summary:**

This paper introduces a implicit Differentiable Out-of-Distribution (OOD) detection layer. This layer addresses outlier detection by solving for fixed points of a differentiable function and using the last iterate of fixed point solver to backpropagate.

**Strengths:**

This is a well-organized and written paper. For example, the language of the introduction section, starting from the statement of the advantage of JFB method to the problem of external knowledge-based multimodal methods, the storyline is smooth, natural, and clear.

The paper is self-contained. Most contributions claimed in the introduction section have the corresponding evidence in the experiment section. Ablation study, analysis, and limitations, all of them be included, organized, and discussed to support the points of this paper.

Finally, such a JFB idea is pretty interesting.

**Weaknesses:**

1. As my understanding, VK-OOD is a method that would not incur many additional parameters. Why is the number of parameters in Table 4 between the baselines and the proposed method have such a big gap (e.g BLIP vs VK-OOD(BLIP), 346M vs 412M)? Does it a fair comparison?

2. This paper did a simple idea, making the iterative algorithm (e.g k-means) end-to-end with networks but making it faster and more efficient by JFB method. I don't know if it is a real problem in that community. Does it really matter? Like the results shown in table 1, basically, the improvement is minor.

This paper is definitely a good paper, I could realize the professionalism of the authors. My concerns are mainly about the contribution. If the authors could offer convincing reasons, I would consider raising my score.

**Questions:**

refer to the weaknesses part.

**Limitations:**

Has been discussed in the main paper, and I agree with that.

---

> ### Author Rebuttal · Authors · 2023-08-10
>
> Thank you for taking the time to read and review our paper. We are glad that you found our paper to be well-written and the proposed JFB for outlier detection idea to be interesting. Please check our general responses for the clarification on the
> model parameters and performance firstly. Then, we will clarify your concerns about the number of parameters and our contributions, please see the responses below:
>
> **Number of parameters - W1:** In VK-OOD, we update $\mu$ and $\sigma$ during the training, which will bring additional parameters. Please refer to the clarification for the number of parameters in the general response 1 and check the updated Table 4 in the rebuttal PDF. As results shown in updated Table 4, the **scalar** VK-OOD also outperforms other baselines with **marginal** increase in the number of parameters, such as VK-OOD-s (BLIP) vs BLIP: 346.4M vs 346M.
>
> **Model performance - W2:** Indeed, Table 1 shows the proof of principles w.r.t time and memory costs of using JFB based OOD feature detection in multimodal pipelines. For the major improvements, please refer to Table 2 and Table 4 to see more details. The results indicate that VK-OOD achieves significant improvements (upto $\approx$ 5 -10 \% increased accuracy)  compared to what baselines achieve in the same settings.
>
> **Our contributions:**
> As noted by other reviewers, we tackle the complex challenging problem of integrating explicit and implicit knowledge in end-to-end multimodal pipelines, aiming to enhance performance while also reducing computational resources. Fortunately, our proposed method is differentiable, has efficiency advantages and can be applied to different datasets and multimodal backbone networks or upstream feature extractors. Furthermore, we provide comprehensive and well-conducted experiments, which illustrate promising understanding and generalization performance improvements on multiple downstream tasks. In short, our main novelty and contribution is the application of JFB method for outlier detection with slightly more parameters in practical multimodal settings.

---

> > ### Comment · Reviewer_MwwQ · 2023-08-17
> > **Feedback by Reviewer MwwQ**
> >
> > Thanks for the response, and I raised my score to borderline accept.

---

### Official Review · Reviewer_h1Px · 2023-07-06

**Soundness:** 3 good
**Presentation:** 3 good
**Contribution:** 3 good
**Rating:** 6
**Confidence:** 3

**Summary:**

The paper presents an approach that combines the features from pre-trained deep networks and freely available semantic explicit knowledge. It proposes an implicit out-of-distribution (OOD) detection layer to address outlier detection and thus further improve understanding and generalization performance in large-scale vision and language settings. It offers comprehensive explanations of the proposed approach, along with experimental results and comparisons against other methods.

**Strengths:**

1. The proposed method is utilizing outlier detection to improve model training, instead of solely for outlier detection.
2. The proposed method is differentiable, efficient and can be applied to different datasets and multimodal backbones.
3. Comprehensive experiments.

**Weaknesses:**

1. The iterative method, though optimized for efficiency, still has concerns in training time.
2. Adding the proposed method sometimes causes degradation as shown in Table 4.

**Questions:**

1. In Table 4, how do you interprete that VK-OOD(BLIP) is worse than BLIP in COCO and Flickr30k (especially yours has more params)?
2. How much more training resource/time were added when adding VK-OOD to other SOTA vision-language models?



**Limitations:**

The iterative method, though optimized for efficiency, still has concerns in training time.

---

> ### Author Rebuttal · Authors · 2023-08-10
>
> Thank you for reviewing our paper. We will address your concerns/questions regarding the training time and model performances as follows:
>
> **Q1:** Please refer to general response 1 and 2, and check the updated Table 4 in the rebuttal PDF. During training, we augmented each captions with 5 external knowledge triplets. Thus, considering the ***noise*** level of textual feature space, the recall drop shown in Table 4 is **marginal**.
>
> **Q2:**
> We pre-train on three datasets with total of 1M images and 6.8M image-caption pairs, which is approximately **30\% less** than what the baselines such as ViLT used in their training. Each caption is augmented with 5 external knowledge triplets. We trained VK-OOD-l (ViLT) on the aforementioned training set with 50k steps on 8 NVIDIA 2080Ti GPU and it took around 2.5 days only. Thus, the overall training time and resources are much more less than ViLT and BLIP. However, our proposed model obtains significant improvements in all downstream tasks with training on less samples. Each sample in VK-OOD-l (ViLT) **only** requires $\approx$ 3ms more than the baseline(ViLT) in the dense case. Our results seem promising in this way!

---

> > ### Comment · Reviewer_h1Px · 2023-08-14
> >
> > Thanks for the reply. I have read the rebuttal.

---

### Official Review · Reviewer_oiir · 2023-07-15

**Soundness:** 3 good
**Presentation:** 3 good
**Contribution:** 3 good
**Rating:** 5
**Confidence:** 4

**Summary:**

This paper introduces a novel approach to integrating explicit knowledge graphs into deep networks for multimodal analysis. To filter noise brought by external knowledge, the authors propose an implicit differentiable Out-of-Distribution (OOD) detection layer with efficient backpropagation. This implicit layer comprises the Expectation Maximization (EM) steps of Gaussian Mixture Models (GMM). To efficiently differentiate through the OOD detection implicit layer, Jacobian-free backpropagation was applied in the final optimization step. The proposed OOD detection layer demonstrates state-of-the-art results with significantly fewer samples and less training time.

**Strengths:**

- The experiment section is well-structured. Clarity of this section is a plus.
- The ablation study is carefully conducted, revealing the influence of hyperparameters and external knowledge resources and the effectiveness of efficient backpropagation.
- Despite having 30% incomplete data, the proposed implicit layer outperforms previous models in the OOD settings. This demonstrates the OOD generalization of implicit layers.
- Informative qualitative results demonstrate how the proposed model performs in the real VQA datasets.

**Weaknesses:**

- Missing references. The current manuscript doesn't effectively integrates previous research into the discussion. The following references are highly relevant and should be acknowledged:
  - Paper [1] is a landmark work in implicit models.
  - Paper [2] proposed the one-step gradient for backpropagating through iterative algorithms, substantially overlapping with and earlier than the efficient differentiation in this work. Both paper [2] and this work leverage non-convex optimization layers of multiple variables.
  - Paper [3] systematically studies the inexact gradient for implicit models in theory and practice.
  - Papers [4,5] delve into the Out-of-Distribution (OOD) generalization of implicit layers and the benefits from path-independence/convergence.
  - Paper [6] discusses the adversarial robustness of implicit models.

- The method section can be rewritten for better clarity. The paper could benefit from an improvement in notation clarity.

Typos:

- Line 314: "saccelerater" should be "accelerate".

[1] Bai, Shaojie, J. Zico Kolter, and Vladlen Koltun. "Deep equilibrium models." Advances in Neural Information Processing Systems 32 (2019).

[2] Geng, Zhengyang, Meng-Hao Guo, Hongxu Chen, Xia Li, Ke Wei, and Zhouchen Lin. "Is Attention Better Than Matrix Decomposition?." In International Conference on Learning Representations. 2020.

[3] Geng, Zhengyang, Xin-Yu Zhang, Shaojie Bai, Yisen Wang, and Zhouchen Lin. "On training implicit models." Advances in Neural Information Processing Systems 34 (2021).

[4] Anil, Cem, Ashwini Pokle, Kaiqu Liang, Johannes Treutlein, Yuhuai Wu, Shaojie Bai, J. Zico Kolter, and Roger B. Grosse. "Path Independent Equilibrium Models Can Better Exploit Test-Time Computation." Advances in Neural Information Processing Systems 35 (2022).

[5] Bai, Shaojie, Zhengyang Geng, Yash Savani, and J. Zico Kolter. "Deep equilibrium optical flow estimation." In Proceedings of the IEEE/CVF Conference on Computer Vision and Pattern Recognition. 2022.

[6] Yang, Zonghan, Tianyu Pang, and Yang Liu. "A Closer Look at the Adversarial Robustness of Deep Equilibrium Models." Advances in Neural Information Processing Systems 35 (2022).

**Questions:**

- How was the OOD detection layer initialized for the EM algorithm? Given that EM may fall into local optima, how does this influence the model's performance in the OOD settings?
- The accuracy gain from iterations doesn't appear to plateau in Figure 3(b). Is the EM algorithm convergent at the maximum iterations? Would it benefit from additional iterations?
- Could you plot the fixed point error $|\mu_{t} - EM(\mu_{t})|$ alongside the optimization steps and include this plot near Figure 3(b)?
- Given the paper's claim of the robustness of the OOD detection layer, I am curious about the adversarial robustness of the proposed VK-OOD in the VQA settings.
- Specifically, could any input perturbations/attacks impact the convergence or stability of the implicit layer? It would be useful for authors to monitor the convergence using the aforementioned fixed point error.

**Limitations:**

The paper discusses the limitations of the proposed OOD detection implicit layer, stating that it could infer inefficiently if the covariance matrix is parameterized densely. A fast linear system solver could potentially rectify this inefficiency. However, these extensions was left for future works.

---

> ### Author Rebuttal · Authors · 2023-08-10
>
> Thank you for taking the time to read and review our paper. We are glad that you found our proposed method with OOD detection implicit layer novel and has efficiency benefits. We have started working on integrating suggested previous related research in the manuscript and will incorporate all the modifications into the revised version. Please check our general responses for the clarification on the model parameters and performance. We now address your  specific questions:
>
> **Q1: How did you initialize your EM algorithm?**
>
> We initialized the OOD detection layer parameters $\mu$ and $\sigma$ using random $k$ number of vision features of the inputs. Our OOD detection layer may fail during inference. However, for training purpose, approximating $\mu^*$ and $\sigma^*$ might be sufficient. With an approximation $\tilde{\mu}$ and $\tilde{\sigma}$, we use the GEM score to filter OOD features in multimodal pipeline. In the experimental results, Table 2 in the paper illustrates that the performance of the model with OOD detection layer is better than the models w/o OOD detection layer in downstream tasks.
>
> **Q2: Is the EM algorithm convergent at the maximum iterations? Would it benefit from additional iterations?**
>
> Please see the general response 3. Perhaps there is a misunderstanding, but we could not think of any reason why accuracy should plateau in Figure 3(b), because the x-axis in Figure 3(b) is the number of triplets/clusters. We think the reviewer may refer to Figure 3(a) which x-axis is $T$ iterations where we can observe plateauing.
>
> Yes, we believe more iterations may be beneficial. Although, considering the training costs, we conduct ablation studies on $T \leq 10$ and the improvements in term of accuracy are slow after $T \geq 5$. Moreover, the main goal of OOD detection layer is to find the approximate clusters for the inputs.
>
>
> **Q3: Can you provide empirical convergence plot of $\mu,\sigma$ of In-Distribution (ID) features using EM algorithm?**
>
> We have provided the fixed point error plot over iterations in the rebuttal pdf. We find that the squared euclidean distance between successive iterates indeed goes to zero. We will add this plot in Figure 3 in the revised version, thank you for the suggestion.
>
> **Q4: How is Adversarial Training related to OOD detection?**
>
> Thank you for the question. We can use one of components $\tilde{\mu}_k$ and $\tilde{\sigma}_k$ to be one of the mixtures to handle adversarial samples.  To calculate this component we can use an appropriate adversarial sample generation or attack algorithm at the feature level. We leave this for future work.
>
> **Q5: Could any input perturbations/attacks impact the convergence or stability of the implicit layer?**
>
> Yes, it  can impact convergence behavior of EM algorithm, and hence our layer. Designing robust outlier detection is a great suggestion and still remains an open problem, so we will consider the problem for future work.

---

> > ### Author Response · Authors · 2023-08-19
> >
> > Dear Reviewer oiir,
> >
> > We would greatly appreciate your feedback on whether our rebuttal has adequately addressed your concerns, and we are more than happy to answer any other remaining questions you may have. We sincerely value your feedback. Many thanks in advance!

---

> > ### Comment · Reviewer_oiir · 2023-08-20
> > **Thanks for response**
> >
> > Thanks for the author's response! It has addressed many of the points I raised. Considering the limited discussion time left, I have first adjusted my score to 5.
> >
> > That said, several concerns still persist. Addressing these is essential prior to the publication of this work. I believe there's room for a deeper exploration into the advantages of convergence and the role of implicit layers, which could elevate the overall quality of the paper.
> >
> > - **Q2**: Sorry for the typo. It should be Figure 3(a). Figure 3(b) in the submitted PDF is nice.
> >
> > - **Q3**: The essence of my third question (and Q2) is how the ID and OOD setup influence the fixed point convergence. Specifically, is there a notable difference in convergence behavior between the two setups? Does it take more iterations to converge in the OOD setup? How does convergence associate with performance in both ID and OOD setups? I did not see a particular figure to compare the ID and OOD convergence. There is only one plot for convergence that illustrates convergence under two gradient methods.
> >
> > - **Q4**: Thanks for replying to this question. Incorporating this discussion as a 'future work' section in the paper would be beneficial.
> >
> > - **Q5**: I understand that implementing attacks can be a bit complex and is likely more suited for future work. However, **assessing a model's generalization through simple input perturbations over a pretrained checkpoint could be feasible in the current scope** and is at an acceptable workload. Introducing Gaussian noise or dropping patches are potential candidates to consider. This request is because we can control the distribution gap by noise levels or drop rates, which can help investigate how the EM implicit layer improves OOD performance. Ideally, I anticipate a plot delineating multiple curves derived from varying noise levels or drop rates, with accuracy plotted on the y-axis against the number of EM steps on the x-axis. Additionally, it would be informative to have a plot illustrating curves from noise levels/drop rates against the convergence $| \mu - EM(\mu) |$, over EM steps. This question extends Figures 3(a) and 3(b) to understand the generalization under the growing distribution gap. If possible, adding a table to the discussion can be helpful. I understand making plots for paper would take some time.

---

> > > ### Author Response · Authors · 2023-08-21
> > > **Thank you for the further discussions**
> > >
> > > Dear Reviewer oiir,
> > >
> > > Thank you for the constructive feedback. We will incorporate more and deeper convergence analysis of our proposed implicit layer in revision, including analysis (and tables/figures) of both ID and OOD setups, and converged rate over optimization steps with different noise levels.
> > >
> > > **Q3:** The OOD samples are not present within the training datasets themselves. Instead, in our proposed method, we encounter outliers when integrating external knowledge triplets into the training pipeline. In other words, if we denote $M$ as the number of external knowledge triplets, then $M=0$ corresponds to the ID setup that you have mentioned. For this ID setup,  we included the model performance using our implicit layer in Table 2 in the paper. Now, we present some quantitative results to answer your question. To delve into this further, we consider two setups: one with $M=0$ i.e., ID setup, and OOD setup with $M=5$ that correspond to augmenting features from external knowledge. We can see from the results in the following table that there is not a significant  difference from the the rate of convergence perspective --- as indicated by squared norm of successive iterates $\|\mu_t-\mu_{t+1}\|_2^2$ --- in both setups viz., ID and OOD setups. However, from the Accuracy column in the following table we conclude that the performance in VQA tasks has significant improvements over iterations in when considering external knowledge, as in the OOD set up with $M=5$.
> > >
> > > |                |          ID(M=0)          |          |          OOD(M=5)         |          |
> > > |:--------------:|:-------------------------:|:--------:|:-------------------------:|:--------:|
> > > | T (iterations) | $\|\mu_t-\mu_{t+1}\|_2^2$ | Accuracy | $\|\mu_t-\mu_{t+1}\|_2^2$ | Accuracy |
> > > |        1       |            1.94           |   73.6   |            2.15           |   73.1   |
> > > |        3       |           0.059           |   73.8   |           0.089           |   74.8   |
> > > |        5       |           0.038           |   73.9   |           0.051           |   76.1   |
> > > |        8       |           0.042           |   73.9   |           0.036           |   76.5   |
> > > |       10       |           0.036           |   74.1   |           0.034           |   76.8   |
> > >
> > > **Q5:** Thank you so much for the suggestion because we have been thinking about it as well! In fact, to evaluate the sensitivity of our model  with respect to the OOD detection performance, we already included some
> > > experiments of incomplete knowledge triplets with missing values in Section 4.1 in the paper. In the implementation of this section, we only dropped language features inputs due to computational reasons -- augmenting image patches requires more resources such as GPU support. However, since $x$ is used for both visual and language features, the implementation remains the same as that of dropping patches in language features. Hence, we assume that it is equivalent to drop either the visual or the language features since the computational effort involved in running EM algorithm depends only on the total number of features. With this assumption, we now present more results of $\|\mu_t-\mu_{t+1}\|_2^2$ and OKVQA performance in term of accuracy over optimization iterations in the table below for easy reference. With higher level of incompleteness, the rate of convergence is slower. However, there is accuracy gain over iterations in all settings.
> > >
> > > |                |    25\% incompleteness    |          |    50\% incompleteness    |          |    75\% incompleteness    |          |
> > > |:--------------:|:-------------------------:|:--------:|:-------------------------:|:--------:|:-------------------------:|:--------:|
> > > | T (iterations) | $\|\mu_t-\mu_{t+1}\|_2^2$ | Accuracy | $\|\mu_t-\mu_{t+1}\|_2^2$ | Accuracy | $\|\mu_t-\mu_{t+1}\|_2^2$ | Accuracy |
> > > |        1       |           2.213           |   48.2   |           2.448           |   47.3   |           2.735           |   44.6   |
> > > |        3       |           0.174           |   48.9   |           0.214           |   47.7   |           0.208           |   45.1   |
> > > |        5       |           0.065           |   50.6   |           0.092           |   48.6   |           0.244           |   46.0   |
> > > |        8       |           0.057           |   51.2   |           0.108           |   49.1   |           0.112           |   46.6   |
> > > |       10       |           0.051           |   51.8   |           0.084           |   49.4   |           0.167           |   46.9   |
> > >
> > > Thanks again for these valuable technical suggestions. We will properly incorporate both of these tables within our plots and the additional discussions in our revision.

---

> > > > ### Comment · Reviewer_oiir · 2023-08-21
> > > >
> > > > Thank you so much for your prompt response! I think it properly answered my question!
> > > >
> > > > This ID and OOD experiments are convincing! Implicit layers' generalization abilities to OOD setups were also explored in prior works [4,5]. Observations in this work further validate the OOD generalization of implicit models and thus contribute to understanding implicit models, especially through adaptive test time computation and the convergence property. For example, through more test time iterations/computation, the convergent performance at 50% incompleteness is able to match fewer iterations at 25% incompleteness. Prior arts [4,5] also demonstrated this ability of DEQ models. This is why these references are related to this work.
> > > >
> > > > These results should be emphasized in the paper. It could further elevate the technical soundness of this work. Now I am in favor of accepting this paper. Thank you again for your efforts and explanation!

---

### Official Review · Reviewer_uxWt · 2023-07-22

**Soundness:** 2 fair
**Presentation:** 2 fair
**Contribution:** 2 fair
**Rating:** 4
**Confidence:** 3

**Summary:**

This paper proposes a framework for multi-modal (text and image) analysis subject to a differentiable framework for out-of-distribution detection in the input space. The method proposes a pipeline to predict the modes of the in-distribution data using a Gaussian mixture model and then leverages it to predict an out-of-distribution (OOD) score for knowledge triplets. These OOD scores are then used to weigh image-text matching as the training objective. The multi-modal encoded features are then used for down-stream applications such as VQA.

**Strengths:**

The paper leverages the GEM [33] out-of-distribution detection (in a differenetiable setup) in the context of text-anomaly-detection, which it then used for image-text score matching.

**Weaknesses:**

While the method seems to have potentials for OOD detection, there are a couple of shortcomings regarding its motivation and its usage for the downstream applications:

- One of the motivations of the paper in the abstract is the difficulty of fine-tuning and deployment as the model size increases. However, compared to other baselines in Table 4, the model has higher number of parameters. I was expecting a model much lighter but with a comparable or slight degraded performance. Compared to simple EM baselines in Table 1, the approach has about 12% faster training time, but has almost similar FLOPS and params (compared to JB-EM). I think the motivation is not well justified for this approach.

- While the OOD detection component should theoretically boost the performance compared to other baselines, in table 4, there is not a big gap in-between methods. By using a ViLT backbone the model gains a reasonable boost compared to the same model (still having more params). Using BLIP backbone this is not the case. The question is whether the OOD component is needed or the performance gain is due to other architectural and training components? Some of these baselines do not seem to be using any OOD component yet they obtain decent/similar results.

- (related to above) It is not clear how much OOD samples exist in the training data and how much it is needed in a model. While the approach seems interesting, it is not clear how much it is useful given the application.

**Questions:**

I went over the paper a couple of times to fully understand it, some parts were not fully clear

- In line 105, x is described as input features including triplets. I am assuming that x only contains the text data and not visual data. Given x_i, mean and standard deviation (std) of the in-distribution (ID) data is computed. Knowing the mean and std directly depends on x_i, are there out-of-distribution data in x_i when computing these values? If so, what's their percentage and how much they can impact the obtained mean and std?

- What is the matching score function m() in Eq. (4)? This is not defined.

- Given the image v_q and triplet l_j similarity score p(v_q, l_j) in Eq. (4), how the final p(v,l) is computed in Eq. (5)? Is it for a specific triplet or some sort of aggregation over all triplets is taken?

- Given the above description my understanding is that the OOD is measured only on the textual data in Eq. (3) of the paper. However, the training objective is to obtain an image-text matching score. Why OOD is computed only on text and not on the joint representation of image and text? Isn't the text-image score m() already capturing the ODD cases which is more relevant for the objective (image-text matching)?

Typos:

- abstract (The following line should be fixed -- remove but): When these models are used for prediction, but they may fail to capture important semantic information and implicit dependencies within datasets.

- start (*) for sigma in Eq. (3) is missing.

**Limitations:**

No concerns

---

> ### Author Rebuttal · Authors · 2023-08-10
>
> Thank you for taking the time to read and review our paper. We would like you to kindly read our general responses for the clarifications on the model parameters and performance. Here we will address your specific concerns and/or questions:
> >One of the motivations of the paper in the abstract is the difficulty of fine-tuning and deployment as the model size increases. However, compared to other baselines in Table 4, the model has higher number of parameters. I was expecting a model much lighter but with a comparable or slight degraded performance. Compared to simple EM baselines in Table 1, the approach has about 12% faster training time, but has almost similar FLOPS and params (compared to JB-EM). I think the motivation is not well justified for this approach.
>
> **What are our motivations?** Our goal is to integrate existing external knowledge into multimodal pipelines. However, upon further investigation, we noticed that external knowledge could be noisy and can hinder training. With this motivation, we developed a differentiable outlier detection layer with decent theoretical and practical advantages. For example, please see the general response 1 and 2, and the rebuttal PDF for the updated Table 4.
>
> >While the OOD detection component should theoretically boost the performance compared to other baselines, in table 4, there is not a big gap in-between methods. By using a ViLT backbone the model gains a reasonable boost compared to the same model (still having more params). Using BLIP backbone this is not the case. The question is whether the OOD component is needed or the performance gain is due to other architectural and training components? Some of these baselines do not seem to be using any OOD component yet they obtain decent/similar results.
>
> **Why there is not a big gap in-between methods?** Please refer to the general response 2. Moreover, as the results shown in Table 2 indicate, both KG and OOD component provide nontrivial gain in understanding and generalization performance in large-scale vision and language settings. Furthermore, we demonstrate the scalability of VK-OOD across different backbone models with diverse architectures and model sizes. Our proposed implicit layer can be regarded as a plug-and-play module for the multimodal pretrained models, making it easy to incorporate with other vision-language models, **as can be seen in our submitted code**.
>
> >It is not clear how much OOD samples exist in the training data and how much it is needed in a model. While the approach seems interesting, it is not clear how much it is useful given the application.
>
> **What are OOD samples and why we need them?**
> The OOD samples are not present within the training datasets themselves. Instead, in our proposed method, we encounter outliers when integrating external knowledge triplets into the training pipeline. For example, when extracting external knowledge triplets using the given captions, certain concepts/objects might be out-of-distribution (OOD) w.r.t textual/visual features present in training datasets. For a detailed illustration of the OOD examples, please refer to Figure 5. In the experimental setup, we perform ablation studies on VK-OOD components as detailed in Section 4.2. As results shown in Table 2, the models with OOD detection components consistently  outperformed those lacking OOD detection layers in both settings.
>
> **Q1: Does $x_i$ include visual features and textual features? What is the fraction of OOD samples?**
>
> Yes! So, $x_i$ represents the union of features from text and image data. Our key idea is to use a mixture model to approximate the distribution of "In-distribution" features as a mixture model parameterized ($\mu_k^*$ and $\sigma_k^*$) using this union of features. Intuitively, if an image/language feature $x_i$ is close to at least one of these components, then they will be considered in-distribution.  In equation (3) we show only score calculation for $l_j$ for simplicity, but we are happy to clarify this under the equation.
>
> In our experiments, we interpret the features from images and captions existing in the training dataset to be In-Distribution (ID) samples, and then we augment the captions from external knowledge which may be OOD samples. In particular, if all external triplets are OOD, the training set will contain $\approx 70\%$ OOD samples at most in the experiments since we augment $\approx5$ external triplets for every textual input.
>
> **Q2:** The matching score function $s$ in equation (4) is the calculation of cosine similarity between image and textual(triplet) features. We will clarify this in the revised version.
>
> **Q3:** The calculation of $p(v_q, l_j)$ is given in equation (4), and then we average all image-triplet pairs, as denoted by Expectation ($\mathbb{E}$) in Eq. 5, during the training.
>
> **Q4: Is it possible to use OOD visual feature in your framework?**
>
> Yes, of course! In our current implementations, we use textual external knowledge resources which are easy to incorporate. Thus, we consider to filter OOD samples from the textual triplets. Our framework can indeed be modified to handle visual OOD features as well, and we will include this feature in our code release. In the future work, we will consider to directly integrate visual knowledge bases, such as Google Images.

---

> > ### Author Response · Authors · 2023-08-19
> >
> > Dear Reviewer uxWt,
> >
> > We would greatly appreciate your feedback on whether our rebuttal has adequately addressed your concerns, and we are more than happy to answer any other remaining questions you may have. We sincerely value your feedback. Many thanks in advance!

---

> ### Comment · Senior_Area_Chairs · 2023-08-21
> **final discussions**
>
> Dear Reviewer,
>
> As discussions come to an end soon, this is a polite reminder to engage with the authors in discussion.
> Please note we take note of unresponsive reviewers.
>
> Best regards,
> \
> SAC

---

### Author Rebuttal · Authors · 2023-08-10

Thank you for taking the time to read and review our paper. We appreciate the reviewers unanimously agree that our submission is sound, well-presented with contributions clearly written. We are committed to fix all the typos and incorporate all other suggested modifications into the revised manuscript. In this response,, we will address all the technical concerns and questions to further strengthen support for our submission.

## 1. Why are the number of parameters higher than the baselines in Table 4? (Reviewer uxWt, h1Px, MwwQ)
To clarify, the parameters ($\mu$ and $\sigma$) of our proposed OOD detection layer can be initialized in two ways: as a **scalar** $\sigma$ or the **dense** matrix where $\sigma$ is a $d \times d$ matrix with d representing the dimension of input embeddings. The number of parameters of OOD detection layer in the scalar case is $d \times k \times \text{batch size} + k \times \text{batch size}$, and the dense case is $d \times k \times \text{batch size} + d \times d \times k \times \text{batch size}$. Note that if we opt for $\sigma$ as a diagonal matrix, each component of the mixture model will require $2d$ parameters, as opposed to just $d$. In our implementation, $d$ is 768, $k$ is 5 and batch size varies depending on different backbones, such as 32 or 64. We have provided an **updated Table 4** with two cases in the rebuttal PDF. Importantly, in the updated Table 4, the number of parameters in scalar VK-OOD are approximately similar to the baselines. Specifically, please consider to compare to other baseline models, while our *scalar* VK-OOD increases the \#parameters slightly -- ***$\approx 0.4$*** million more parameters, it **significantly** *improves* the performance in many of the downstream tasks that we considered in our submission.

## 2. What improvements does VK-OOD demonstrate in Tables 1 and 4? (Reviewer uxWt, h1Px, MwwQ)
In **updated** Table 1 in the rebuttal pdf, we highlight the empirical advantages (e.g., computational efficiency) of the Jacobian Free Backpropagation (JFB) method in multimodal settings compared to alternative gradient calculation methods. The results illustrate that the proposed  JFB based implicit layer is beneficial to reduce gradient computation costs in term of time and memory usage.

Reviewers will be able to see that **updated**  Table 4 in rebuttal pdf clearly shows model performance improvements in term of appropriate evaluation metrics on specific downstream tasks as we explain as follows:
- For the visual understanding tasks (such as VQA and NLVR), both our scalar and dense VK-OOD seem beneficial. For example, in NLVR2, our dense VK-OOD achieves the best result in terms of accuracy with **10\% and 1.7\% increase** than ViLT and BLIP respectively. The scalar VK-OOD also outperforms all baselines with similar number of parameters in the understanding tasks shown in the updated Table 4.
- For the retrieval tasks, the benefits of using VK-OOD is not so significant, and but shows the robustness of the proposed implicit layer for filtering features. We will explain why this empirical result is natural. Please recall that in the experiments, the goal in retrieval is to retrieve images from text queries or to retrieve texts from images. Now, since we augment captions with external knowledge which are known to induce ***noisy textual features***, the slight improvement in image/text retrieval tasks in COCO and Flickr30k dataset is expected. The numbers reported by the baseline have no such external knowledge, indicating that our implementation works as intended! Finally, even in the retrieval tasks, it may be possible to improve performance by adjusting the noise levels (number of external knowledge triplets) during the fine-tuning process.

## 3. Is it necessary to calculate the fixed point exactly for training in multimodal pipeline? Is it ok to use good EM with few iteration for clustering multimodal features? (Reviewer oiir, NYTC)

No, for training purpose, an approximate $\mu^*$ and $\sigma^*$ is sufficient. In VK-OOD training pipeline, we only require that the gradients provided by the proposed OOD detection layer to be ***a descent direction** with respect to the loss as a function of network parameters*. Thus, as long as the gradient backpropagated is correlated (or has a nonnegative inner product) with the true gradient of upstream layers, the updated weights will still *decrease* the loss function, thereby making optimization progress. Formally, by nonnegative correlation, we need the gradient calculated to satisfy: $\left(g_{\mu^*,\sigma^*}\right)^{\top} g_{\tilde{\mu},\tilde{\sigma}} \geq 0$, where $\tilde{\mu}, \tilde{\sigma}$ is an approximation returned by finite iterations of EM algorithm of the true clusters $\mu^*,\sigma^*$.

---

### Decision · Program_Chairs · 2023-09-21

**Decision:**

Accept (poster)

**Comment:**

The paper focuses on the setting of multi-modal learning, where external expert sources are combined with clean data. The idea is that the expert sources might not always be clean and outliers must be detected and discarded. Outlier detection is then described as an Out-of-Distribution layer. Outliers then are discovered by solving a fixed-point problem, and employing differentiable solvers for backpropagation.

Reviewers are on the borderline but basically positive. They all appreciate the novelty and added value, and negative criticisms are with respect to whether outliers are indeed discovered. However, the authors provide additional experiments to prove so. All in all, this is a submission where reviewers are favorable and I recommend acceptance.